# *Para*-Halogenation of Amphetamine and Methcathinone Increases the Mitochondrial Toxicity in Undifferentiated and Differentiated SH-SY5Y Cells

**DOI:** 10.3390/ijms21082841

**Published:** 2020-04-18

**Authors:** Xun Zhou, Jamal Bouitbir, Matthias E. Liechti, Stephan Krähenbühl, Riccardo V. Mancuso

**Affiliations:** 1Division of Clinical Pharmacology & Toxicology, University Hospital Basel, 4031 Basel, Switzerland; xun.zhou@unibas.ch (X.Z.); jamal.bouitbir@unibas.ch (J.B.); matthias.liechti@usb.ch (M.E.L.); riccardo.mancuso@unibas.ch (R.V.M.); 2Department of Biomedicine, University of Basel, 4031 Basel, Switzerland; 3Swiss Centre for Applied Human Toxicology, 4031 Basel, Switzerland

**Keywords:** amphetamine, methcathinone, mitochondria, neurotoxicity, *para*-halogenation

## Abstract

Halogenation of amphetamines and methcathinones has become a common method to obtain novel psychoactive substances (NPS) also called “legal highs”. The *para*-halogenated derivatives of amphetamine and methcathinone are available over the internet and have entered the illicit drug market but studies on their potential neurotoxic effects are rare. The primary aim of this study was to explore the neurotoxicity of amphetamine, methcathinone and their *para*-halogenated derivatives 4-fluoroamphetamine (4-FA), 4-chloroamphetamine (PCA), 4-fluoromethcathinone (4-FMC), and 4-chloromethcathinone (4-CMC) in undifferentiated and differentiated SH-SY5Y cells. We found that 4-FA, PCA, and 4-CMC were cytotoxic (decrease in cellular ATP and plasma membrane damage) for both cell types, whereby differentiated cells were less sensitive. IC_50_ values for cellular ATP depletion were in the range of 1.4 mM for 4-FA, 0.4 mM for PCA and 1.4 mM for 4-CMC. The rank of cytotoxicity observed for the *para*-substituents was chloride > fluoride > hydrogen for both amphetamines and cathinones. Each of 4-FA, PCA and 4-CMC decreased the mitochondrial membrane potential in both cell types, and PCA and 4-CMC impaired the function of the electron transport chain of mitochondria in SH-SY5Y cells. 4-FA, PCA, and 4-CMC increased the ROS level and PCA and 4-CMC induced apoptosis by the endogenous pathway. In conclusion, *para-*halogenation of amphetamine and methcathinone increases their neurotoxic properties due to the impairment of mitochondrial function and induction of apoptosis. Although the cytotoxic concentrations were higher than those needed for pharmacological activity, the current findings may be important regarding the uncontrolled recreational use of these compounds.

## 1. Introduction

Amphetamine is a potent central nervous system (CNS) stimulant, which is or has been used for the treatment of attention deficit hyperactivity disorder (ADHD), narcolepsy, and body weight control under restricted medical prescription [1]. The predominant pharmacological action of amphetamine is to promote the release from [2] and inhibition of the reuptake of catecholamines into presynaptic nerve terminals [3], causing an increase in the catecholamine concentration in the synaptic clefts of mainly dopaminergic and noradrenergic neurons. A series of structurally similar psychoactive substances were derived from amphetamine after 1877 when it was first synthesized. Methcathinone, an amphetamine derivative, is currently used widely as a recreational drug [4]. Since amphetamine and methcathinone are illicit for recreational use in most countries, many derivatives of these compounds were synthesized as novel psychoactive substances (NPS) and labeled “legal highs” [5]. Halogenation has been recognized as a possible method to create “legal highs” form amphetamine and derivatives [6], and as a method to enhance their membrane binding and permeation characteristics [7]. Many reports have documented how halogenated amphetamine derivatives such as 4-chloroamphetamine (PCA, also called *para*-chloroamphetamine; 4-CA), 4-fluoroamphetamine (4-FA), and methcathinone derivatives such as 4-chloromethcathinone (4-CMC) and 4-fluoromethcathinone (4-FMC) have lately reached the market (Appendix A) [8,9,10,11,12].

However, halogenation also has the potential to increase the toxicity of the original compounds [13]. It is well known that the brain, skeletal muscle, and the liver are the most important target organs of the toxicity of psychoactive substances [14]. Accordingly, the use of amphetamines and methcathinones can lead to the impairment of cognitive function, addiction, convulsions, rhabdomyolysis and acute liver failure [15,16,17]. Our previous research has shown that the *para*-halogenated derivatives of amphetamine and methcathinone caused hepatotoxicity [18] and myotoxicity [19] mainly by impairing mitochondrial function due to disruption of the mitochondrial respiratory chain, which is associated with an increase in the intracellular level of reactive oxygen species (ROS).

To date, systematic toxicological investigations of the *para-*halogenated derivatives of amphetamine and methcathinone in human neuron-like cells are missing. Based on these considerations, the aim of the current study was to investigate the in vitro mechanisms causing neurotoxicity of amphetamine, 4-FA, PCA, methcathinone, 4-FMC, and 4-CMC, using the well-established SH-SY5Y cell model in the differentiated and undifferentiated states [20,21,22].

## 2. Results

### 2.1. Cell Differentiation

As the first step in our study, we validated our protocol of differentiation of SH-SY5Y cells by immunohistofluorescence (Appendix A). Microtubule-associated protein 2 (MAP2) and neurofilament heavy polypeptide (NF-H) are well-known markers of differentiation of SH-SY5Y cells [20]. In undifferentiated SH-SY5Y cells, MAP2 was distributed strongly within the perinuclear area, but weakly in the neurites, and the expression of NF-H was restricted mainly to the cell soma (Appendix A) [20]. Moreover, the neuritic processes were short and almost undetectable in undifferentiated cells. A different pattern of immunoreactivity and cell morphology was observed in ATRA/BDNF-differentiated SH-SY5Y cells (Appendix A). MAP2 and NF-H expression was not only restricted to the cell soma but strongly present in the neurites as well, indicating the typical differentiation of SH-SY5Y cells to neuron-like cells. Finally, differentiated SH-SY5Y cells showed phenotypical characteristics of neurons, in particular a pyramidal body shape with longer projections, similar to dendrites and axons [22].

### 2.2. Cell Membrane Integrity and ATP Content

The release of adenylate kinase (AK) was measured as a marker for cell membrane integrity, and the intracellular ATP content was assessed to determine mitochondrial function (Figure 1 and Figure 2, respectively). Undifferentiated and differentiated SH-SY5Y cells were treated with increasing concentrations of amphetamine, 4-FA, PCA, methcathinone, 4-FMC, and 4-CMC (see Appendix A for the chemical structures of these compounds). All of these compounds were membrane toxic and decreased the intracellular ATP content in a concentration-dependent manner, with the exception of methcathinone and 4-FMC, which did not show any toxicity up to 2000 μM.

The exposure of undifferentiated SH-SY5Y to amphetamine, 4-FA, and 4-CMC for 24 h was significantly membrane toxic at 2000 μM, whereas PCA was toxic already at 500 μM (Figure 1A,C). A similar pattern was observed in differentiated cells for 4-FA, PCA, and 4-CMC, but not for amphetamine (Figure 1B,D). Moreover, membrane toxicity of PCA in differentiated cells started already at 200 μM (Figure 1A,B). The intracellular ATP content started to decrease in undifferentiated cells at 200 μM for PCA, at 500 µM for 4-FA and 4-CMC, and at 1000 μM for amphetamine, whereas methcathinone and 4-FMC were not toxic up to 2000 μM (Figure 2A,C). In differentiated cells, only PCA (starting at 500 µM), 4-FA (starting at 1000 µM) and 4-CMC (starting at 1000 µM) were toxic, whereas amphetamine, methcathinone and 4-FMC did not significantly decrease the cellular ATP pool (Figure 2C,D). Table 1 presents the IC_50_ values of cellular membrane toxicity and ATP depletion for the compounds investigated. The values show a more pronounced effect regarding the diminution of the cellular ATP content as compared to the membrane toxicity, a pattern suggesting mitochondrial toxicity.

### 2.3. Mitochondrial Membrane Potential

In order to confirm mitochondrial toxicity, we determined the mitochondrial membrane potential (Δψ_m_) using JC-1 staining. Mitochondrial toxicants have been shown to decrease the mitochondrial membrane potential in SH-SY5Y cells [23]. Each of 4-FA, PCA and 4-CMC decreased the Δψ_m_ in a concentration-dependent manner in both cell models (Figure 3A–D), whereas amphetamine, methcathinone and 4-FMC were not toxic up to 2000 µM. Similar to cytotoxicity, PCA was the most toxic compound investigated, starting to decrease Δψ_m_ at 500 µM in undifferentiated and at 200 µM in differentiated SH-SY5Y cells exposed for 24 h (Figure 3A,B). In comparison to the amphetamines, the methcathinones were less toxic.

These data confirmed our results obtained for the depletion of the cellular ATP content and indicated that 4-FA, PCA and 4-CMC are mitochondrial toxicants.

### 2.4. Cellular Oxygen Consumption

In order to understand the mechanism of mitochondrial toxicity, we determined the cellular oxygen consumption by SH-SY5Y cells having been exposed for 24 h to the test compounds using a Seahorse XF96 analyzer. After having obtained the basal respiration, oligomycin was injected to inhibit complex V allowing the determination of the leak respiration as a measure of the uncoupling of oxidative phosphorylation. This was followed by the addition of FCCP to obtain the maximal respiration and the complex I inhibitor rotenone to obtain the non-mitochondrial respiration. Basal respiration, leak respiration and maximal respiration are shown in Figure 4. As can be seen in all Figures (Figure 4A–L), FCCP did not stimulate basal respiration in differentiated and even inhibited basal respiration in undifferentiated cells. None of the drugs stimulated the leak respiration, suggesting that there was no uncoupling of oxidative phosphorylation. PCA showed a concentration-dependent effect on basal respiration and respiration in the presence of FCCP in differentiated cells, reaching statistical significance at 100 µM (Figure 4F). In undifferentiated cells, the effect of PCA was not clearly concentration-dependent but reached statistical significance already at 50 µM (Figure 4E). Similarly, 4-CMC was toxic for undifferentiated and differentiated SH-SY5Y cells, reaching statistical significance for basal and stimulated respiration at 500 µM in undifferentiated (Figure 4K) and at 200 µM in differentiated SH-SY5Y cells (Figure 4L). A trend to decrease basal and stimulated respiration in both cell types with 4-FMC was shown, but without reaching significance up to 1000 µM (Figure 4I,J). In comparison, amphetamine, 4-FA, and methcathinone did not affect the oxidative metabolism of SH-SY5Y cells.

These data confirmed mitochondrial toxicity of PCA and 4-CMC, but not of 4-FA. They show that halogenation is critical for the toxicity of these compounds and that the addition of chlorine in the *p*-position is more toxic than fluoride.

### 2.5. Mitochondrial Superoxide Production

Toxicants inhibiting the function of the mitochondrial electron transport chain can increase the production of mitochondrial ROS [19,24]. Therefore, we determined mitochondrial production of the superoxide anion in undifferentiated and differentiated SH-SY5Y cells exposed to test drugs for 24 h (Figure 5).

In both undifferentiated and differentiated SH-SY5Y cells exposed to 4-FA, mitochondrial superoxide production started to increase at the highest concentration of 2000 μM (Figure 5A,B). Concerning PCA, only undifferentiated SH-SY5Y cells showed a significant increase of the cellular ROS content, which started at 500 μM (Figure 5A), whereas the differentiated cells were more resistant with a trend for an increase at 2000 µM (Figure 5B). The exposure to 4-CMC increased the superoxide anion content in both cell models starting at 2000 μM (Figure 5C,D). The other drugs investigated did not increase the mitochondrial superoxide production in the investigated concentration range (up to 2000 µM).

### 2.6. Mechanisms of Cell Death

When the mitochondrial damage becomes too large, cells undergo apoptosis or necrosis, depending on the cellular ATP content [25]. In order to investigate the mechanism of cell death, we assessed the externalization of phosphatidylserine by annexin V binding, and the permeability of the cell membrane to propidium iodide (PI), as markers of apoptosis and necrosis, respectively. H_2_O_2_ (500 µM) was used as a positive control (Figure 6) [26]. In undifferentiated SH-SY5Y cells exposed for 6 h, PCA increased necrosis starting at 200 µM, reaching statistical significance at 500 µM (Figure 6A). 4-CMC induced apoptosis starting at 1000 µM and reached statistical significance at 2000 µM. Similar to PCA, 4-FA also increased necrosis (starting at 2000 µM), whereas the other compounds did not induce apoptosis or necrosis up to 2000 µM. In differentiated SH-SY5Y cells, PCA induced necrosis starting at 100 µM, reaching statistical significance at 500 µM (Figure 6B). 4-CMC induced apoptosis starting at 500 µM, reaching statistical significance at 1000 µM, and necrosis starting at 2000 µM. 4-FA induced necrosis starting at 2000 µM, whereas the other compounds were ineffective. The shorter incubation time (6 h versus 24 h) was selected to focus on the early apoptosis phase rather than on necrosis.

To confirm the activation of apoptotic pathways and to find out the mechanism of activation, we assessed the activation of caspases 3 and 9 in the presence of 4-CMC at 6 h of incubation. We chose 4-CMC for these experiments, since only 4-CMC, and not PCA or 4-FA, induced apoptosis of SH-SY5Y cells at 6 h of incubation. Caspase 9 is activated in response to the intrinsic apoptotic pathway, whereas caspase 3 is an executioner caspase [27,28]. The intrinsic pathway can be defined as the mitochondrial pathway, since damaged mitochondria release cytochrome *c*, which activates caspase 9 [29]. To confirm that 4-CMC causes apoptosis and to assess by which pathway, we determined the expression of cleaved caspase 3 and 9 by immunoblotting after drug exposure for 6 h (Figure 7A,B) [30]. Amiodarone (50 µM) and MDMA (500 µM) were used as negative controls regarding the induction of apoptosis. As shown in Figure 7A, 2000 μM 4-CMC increased the abundance of cleaved caspase 3 in both undifferentiated and differentiated SH-SY5Y cells, confirming the results obtained with the annexin V assay. Moreover, 4-CMC increased the expression of cleaved caspase 9 at 2000 µM in both differentiated and undifferentiated SH-SY5Y cells (Figure 7B).

## 3. Discussion

We found that both halogenated amphetamines were cytotoxic (membrane damage and/or ATP depletion) for undifferentiated and differentiated SH-SY5Y cells, with a similar concentration-dependent pattern. Among the halogenated methcathinones, only 4-CMC showed plasma membrane toxicity and ATP depletion. Amphetamine was cytotoxic at the highest concentration investigated, whereas methcathinone was not cytotoxic up to 2 mM. The rank of cytotoxicity observed for the *para*-substituents was chloride > fluoride > hydrogen for both amphetamines and cathinones. Both PCA and 4-CMC impaired the function of the mitochondrial electron transport chain and increased mitochondrial superoxide production. In general, differentiated SH-SY5Y were less sensitive to the chemicals investigated than undifferentiated cells.

The amphiphilic nature of amphetamines and methcathinones enables them to cross the blood-brain barrier and to reach the neurons, their site of pharmacological action. As known from case reports and as shown in the current study, this has not only an impact on the pharmacological effect but also on the neurotoxicity of these compounds. Halogenation in the *para*-position not only increases lipophilicity of these chemicals but also blocks *para*-hydroxylation, prolonging the pharmacological and toxicological effects of these compounds in vivo [31].

In the current investigations, 4-FA, PCA and 4-CMC decreased the cellular ATP content at lower concentrations than impairing membrane integrity (Table 1 and Appendix A). This is a typical feature of toxicants that impair mitochondrial function [32,33] and is in line with our previous investigations on the hepatocellular toxicity of amphetamine, methcathinone and their *para*-halogenated forms [3,13,34,35]. A decrease in the cellular ATP content is also one of the first hints of mitochondrial perturbation, since mitochondria are the main site of ATP production in most cells, although glycolysis may also contribute when glucose is available [36]. The significant drop in the Δψ_m_, which appeared at similar concentrations as the decrease in the cellular ATP pool (Appendix A), confirmed that mitochondria are the target of the *p*-halogenated amphetamine compounds and of 4-CMC [37]. These findings are consistent with the observations of Chen et al., who investigated the toxicity of amphetamines in a pulmonary artery model [18,38] and of Luethi et al., who studied the mechanisms of hepatocellular toxicity associated with new synthetic cathinones [18,38]. Investigations of the effects on mitochondrial respiration revealed that PCA and 4-CMC reduced basal and maximal respiration of both undifferentiated and differentiated SH-SY5Y cells, whereby PCA was slightly more toxic than 4-CMC. While amphetamine and methcathinone were not toxic up to 2000 µM, the fluorinated derivatives decreased basal and maximal respiration by trend in both cell types studied. The toxicity rank order of the substituents in the *p*-position for disturbing mitochondrial function was therefore for both amphetamines and methcathinones (Cl > F > H), which was also the case for cytotoxicity. The current study revealed a decrease in oxygen consumption by SH-SY5Y cells in the presence of 4-FA, PCA and 4-CMC, indicating impaired function of the electron transport chain. The results do not, however, allow to conclude which enzyme complexes of the electron transport chain are affected. In a previous study, we have shown that the amphetamine-like MDMA reduced the activity of complexes I and III, and the cathinones α-pyrrolidinopentiophenone (α-PVP) and naphyrone complex I and complex II, respectively, in C2C12 cells, a mouse skeletal muscle cell line [19]. Inhibition of complex I and/or complex III could explain the results obtained in the current study.

Defective mitochondrial respiration can lead to an increase in mitochondrial ROS levels [3,39]. It is known that the neurotoxicity of amphetamines is at least partially due to an increased cellular ROS accumulation caused by oxidative deamination of catecholamines, depletion of antioxidant systems and/or mitochondrial dysfunction [24,40,41,42,43,44]. In addition, Naserzadeh et al. have shown that 4-methylmethcathinone (4-MMC), a new and popular drug of abuse, increases mitochondrial ROS levels and damages the outer mitochondrial membrane in mitochondria obtained from the hippocampus, cortex, and cerebellum of rats [45]. In the current study, we observed a significant increase in the mitochondrial ROS content in the presence of 4-FA, PCA and 4-CMC in undifferentiated SH-SY5Y cells, which is in line with the studies mentioned above.

Mitochondrial ROS generation can have different consequences for cells, eventually leading to oxidation of DNA, membrane lipids and proteins and finally cell death. Superoxide generated mainly by complex I or III within the mitochondria can be degraded by superoxide dismutase 2 (SOD2) to H_2_O_2_, which can leave the mitochondria and oxidize KEAP1 in the cytoplasm [46]. This process stabilizes and activates NRF2, a nuclear transcription factor, which stimulates the transcription of genes associated with antioxidative defense, such as SOD2 and many others [47]. An additional defense mechanism that can be activated by ROS-induced accumulation of misfolded proteins and/or organelle damage is autophagy [48]. We have shown in a recent study that 4-CMC stimulates autophagy in SH-SY5Y cells [49], demonstrating the activation of antioxidative defense mechanisms. If mitochondrial superoxide generation is too extensive, mitochondrial superoxide accumulation can lead to an opening of the permeability transition pore (mPTP), which is associated with a drop in the mitochondrial membrane potential and release of cytochrome *c* into the cytoplasm with induction of apoptosis [50,51]. This is in line with the activation of caspases 9 and 3 observed in the current study, which proves the stimulation of the endogenous apoptotic pathway by 4-CMC.

When compared to differentiated cells, the effects of PCA and 4-CMC were less accentuated than in undifferentiated cells and the effect of 4-FA was not significant. In agreement with these findings, Schneider et al. showed that the regulation of glycolysis and oxidative phosphorylation is modulated during cell differentiation, which can affect the way cells respond to toxicants causing oxidative stress [52]. In line with this notion, ATRA-induced cell differentiation has been shown to confer resistance to compounds inducing oxidative stress [53]. However, despite an increase in antioxidative capacity during the differentiation of SH-SY5Y cells [20,22,24], treatment with PCA and 4-CMC caused high enough cellular ROS levels to induce apoptosis.

There are reports in the literature that show how the use of methamphetamine is linked to the development of Parkinson’s syndrome. In an epidemiological study, an increased risk to develop Parkinson’s syndrome by a factor of 1.5–3 has been reported in amphetamine consumers [54,55]. Parkinson’s syndrome develops due to a loss of dopaminergic neurons in the nigrostriatal system, which may be caused by mitochondrial damage [56]. Therefore, the current study suggests that *para-*halogenation of amphetamines and methcathinones may increase this risk. If confirmed, this is an important finding, which may have an impact on the use of these compounds.

In comparison to their pharmacological activity, which is observed in the high nanomolar to low micromolar range, depending on the compound and the pharmacological effect considered [3], cytotoxicity was detected at clearly higher concentrations in the current study. Plasma concentrations reached for amphetamine after a pharmacological dose are in the low micromolar range [57] and PCA started to be toxic at 50 to 100 micromolar. A possible explanation for this discrepancy between pharmacological activity and toxicity may be that cell lines as used in the current study may be less sensitive to toxicants than primary cells. This has for instance been shown for human hepatocyte cell lines and primary human hepatocytes [58]. Furthermore, patients presenting with neurotoxicity usually have ingested higher than pharmacological doses of these compounds, may have ingested other toxic drugs and/or alcohol and may have an elevated body temperature. It has recently been shown that the hepatocellular toxicity of the synthetic cathinone 3,4-methylenedioxypyrovalerone is more accentuated at higher temperatures [59]. In addition, since amphetamine has a volume of distribution in the range of 3 L/kg [57], it can be assumed that the concentrations in the brain are higher than those in plasma. Taken together, these factors may be sufficient to explain the gap between the concentrations associated with a pharmacological activity and toxic effects observed for these compounds in the current in vitro investigations.

## 4. Materials and Methods

### 4.1. Chemicals

Amphetamine, 4-fluoroamphetamine (4-FA), methcathinone, 4-fluoromethcathinone (4-FMC), 4-chloromethcathinone (4-CMC), and 3,4-methylenedioxymethamphetamine (MDMA) were purchased from Lipomed (Arlesheim, Switzerland). 4-Chloroamphetamine (PCA) was purchased from Cayman Chemical (Ann Arbor, MI, USA). All drugs were racemic hydrochloride salts with a HPLC purity of >98%. Test drugs were dissolved in DMSO and stored at -20 °C, then serially diluted in DMSO to avoid precipitation, followed by dilution in medium or assay buffer. The final DMSO concentration during the experiment was 0.1%.

### 4.2. Cell Culture and Differentiation

SH-SY5Y cells were obtained from the European Collection of Authenticated Cell Cultures (ECACC, RRID:CVCL_0019) (Sigma-Aldrich, Buchs, Switzerland). Undifferentiated SH-SY5Y cells were cultured in high glucose Dulbecco’s Modified Eagle’s Medium (DMEM) (Thermo Fischer Scientific, Basel, Switzerland) supplemented with 15% heat-inactivated fetal bovine serum (FBS) (Thermo Fischer Scientific, Basel, Switzerland), 2 mM L-glutamine (Thermo Fischer Scientific, Basel, Switzerland), and 1 mM sodium pyruvate (Thermo Fischer Scientific, Basel, Switzerland) at 37 °C in a humidified 5% CO_2_ atmosphere. Confluent cells (70–80%) were passaged using Gibco™ Trypsin-EDTA (0.05%) reagent (Invitrogen, Basel, Switzerland). Neuron-like differentiation of SH-SY5Y cells were induced as described previously [20]. Briefly, SH-SY5Y cells were seeded into collagen-coated plates (Thermo Fischer Scientific, Basel, Switzerland) at 10,000 cells/cm^2^. Following overnight growth, the medium was replaced by culture medium supplemented with 10 μM all-trans-retinoic acid (ATRA) (Sigma-Aldrich, Buchs, Switzerland) and the plates were incubated for five days. Finally, medium was replaced by DMEM supplemented with 50 ng/mL brain-derived neurotrophic factor (BDNF) (Sigma-Aldrich, Buchs, Switzerland) and incubated for 7 days [20]. To avoid phenotypic alterations, differentiated SH-SY5Y cells were used between the 14th and 16th passage [22,60].

### 4.3. Cell Differentiation Determined by Microscopy

To validate the differentiation protocol, we performed immunofluorescence experiments in order to check cell morphology and the expression of differentiation markers. Undifferentiated SH-SY5Y cells were seeded into ibiTreat µ-Slide (Vitaris, Baar, Switzerland) one day before the assessment. Differentiated SH-SY5Y cells were grown into collagen-coated µ-Slide (Vitaris, Baar, Switzerland). For the staining procedure, cells were fixed with 100 μL of fixation buffer (3.7% paraformaldehyde in PBS) for 20 min at room temperature (RT). Then, cell permeabilization was achieved by adding 0.1% Triton^®^ X-100 in PBS solution for 10 min. Finally, the cells were blocked for 20 min in blocking solution (10% goat serum, 0.1% Triton^®^ X-100 in PBS). Afterwards, the primary antibody cocktail (anti-MAP2 clone Poly18406, RRID:AB_256545, diluted 1:500, anti-neurofilament H clone SMI31, RRID:AB_2566782, diluted 1:200 in blocking solution) (BioLegend, San Diego, CA, USA) was added at RT for 1 h. Then, the cells were washed with PBS and the secondary antibody cocktail (FITC anti-rabbit clone Poly4064, RRID:AB_893531, diluted 1:250, Alexa Fluor^®^ 647 Goat anti-mouse IgG polyclonal, RRID:AB_2563045, diluted 1:100, and 1.8 μM Hoechst 33258 in blocking solution) (BioLegend, San Diego, CA, USA) was added to each well and the slide was incubated at RT for 1 h with light protection [20]. Samples were maintained in PBS and investigated using an Olympus IX83 microscope (Olympus, Shinjuku, Japan).

### 4.4. Cell Membrane Integrity

Membrane toxicity was assessed by measuring adenylate kinase (AK) release using the ToxiLight Bioassay kit (Lonza, Basel, Switzerland) according to the manufacturer’s protocol [19]. Briefly, undifferentiated SH-SY5Y (50,000 cells/well) and differentiated SH-SY5Y (25,000 cells/well) cells were exposed to different concentrations of amphetamine, 4-FA, PCA, methcathinone, 4-FMC, and 4-CMC (100 µM, 200 µM, 500 µM, 1000 µM, and 2000 µM). Triton^®^ X-100 (0.1%) was used as a positive control to induce cell lysis. After 24 h of exposure, 20 μL of cell supernatant was transferred into a luminescence compatible 96-well plate and 50 μL of AK detection reagent was added to each well. The plate was incubated for 5 min at RT. The luminescence was measured on an M200 Pro Infinity plate reader (Tecan, Männedorf, Switzerland). The percentage of intact cells (no cell membrane integrity loss) was calculated in relation to DMSO-treated and Triton^®^ X-100 treated cells, representing 100% and 0% of intact cells, respectively.

### 4.5. ATP Content

The intracellular ATP content was measured using the CellTiter-Glo^®^ kit (Promega, Dübendorf, Switzerland) according to the manufacturer’s protocol [19]. Undifferentiated and differentiated neuronal SH-SY5Y cells were prepared as described above. Briefly, 80 μL assay buffer was added to each 96-well containing 80 μL culture medium. After 15 min of incubation at RT, the ATP content was determined by luminescence measurement using an M200 Pro Infinity plate reader (Tecan, Männedorf, Switzerland)**.** All data were normalized to control incubations containing DMSO 0.1%.

### 4.6. Mitochondrial Membrane Potential

The mitochondrial membrane potential (Δψ_m_) was determined using the cationic dye 5,5,6,6-tetrachloro-1,1,3,3-tetraethylbenzimidazolylcarbocyanine iodide (JC-1) kit (Abcam, Cambridge, UK) according to the manufacturer’s protocol [23]. Undifferentiated and differentiated SH-SY5Y cells were seeded as described above (cell membrane integrity assay) in black costar 96-well plates and exposed to test drugs for 24 h. Carbonyl cyanide-*p*-trifluoromethoxyphenylhydrazone (FCCP, 100 μM) was used as a positive control. FCCP is an uncoupler of mitochondrial oxidative phosphorylation and therefore decreases Δψ_m_ [23]. FCCP was added to the cells for 4 h. The medium was removed and the cells were washed with 100 μL/well of 1X dilution buffer from the JC-1 kit. The working JC-1 solution (20 μM JC-1 in 1X Dilution Buffer) was freshly prepared and 100 μL was added into each well. The plate was incubated for 10 min at 37 °C with light protection. Then, the wells were washed twice with 1X dilution buffer solution. The fluorescence was measured by an M200 Infinite Pro plate reader (Tecan, Männedorf, Switzerland) at an excitation wavelength of 475 nm for both aggregate and monomer forms. The emission wavelength was set at 530 nm and at 590 nm for monomer and aggregate forms, respectively. The ratio of the fluorescence intensities between aggregates and monomers was considered as Δψ_m_. All data were normalized to control incubations containing DMSO 0.1%.

### 4.7. Oxygen Consumption

To measure the changes in mitochondrial respiration after test drug exposure, the mitochondrial oxygen consumption rate (OCR) was measured with a Seahorse XF96 analyzer (Seahorse Biosciences, North Billerica, MA, USA). Undifferentiated and differentiated SH-SY5Y cells were cultured in XF96 Cell Culture Microplates (Seahorse Biosciences, North Billerica, MA, USA), which were pre-coated with the cell adhesive Corning™ Cell-Tak (22.4 μg/mL, Corning, New York, NY, USA). The attached cells were treated with culture medium containing the test drugs (PCA at a concentration of 50–200 μM and the remaining drugs at concentrations of 200–100 μM) for 24 h. Upon treatment, the culture medium was replaced with 175 μL per well of unbuffered medium (4 mM L-glutamate, 1 mM pyruvate, 1 g/L glucose, and 63.3 mM sodium chloride, pH 7.4). Thereafter, the cells were equilibrated for 40 min in a CO_2_-free incubator at 37 °C, and the plate was transferred into the XF96 analyzer. Basal oxygen consumption rates were measured prior to the automated injection of an inhibitor of F_0_F_1_ATPase (oligomycin 1 μM). In order to assess the maximal respiratory capacity, 1 μM FCCP was applied, which uncouples the activity of the electron transport chain from ATP synthesis. Finally, followed by the addition of an electron transport chain complex I inhibitor (rotenone 1 μM), the non-mitochondrial respiration rate was obtained [61]. OCR was automatically recorded by the Wave software (Seahorse Biosciences, North Billerica, MA, USA), and data were normalized to protein content. The protein content was determined using the Pierce BCA Protein Assay kit (Thermo Fisher Scientific, Basel, Switzerland). OCR was expressed as pmol O_2_ per minute per mg of protein.

### 4.8. Mitochondrial Superoxide Production

Mitochondrial superoxide production was determined using the MitoSOX™ Red fluorophore probe (Thermo Fisher Scientific, Basel, Switzerland). MitoSOX is a living-cell permeant fluorogenic dye commonly used for the detection of superoxide within mitochondria. In brief, undifferentiated and differentiated SH-SY5Y cells were cultured and treated in black clear-bottom 96-well plates [3]. Amiodarone (50 μM) was used as a positive control [62]. The cells were treated with the test drugs at concentrations of 100–2000 μM for 24 h at 37 °C in a CO_2_ incubator and then washed twice with PBS. Subsequently, the medium was replaced by 100 μL PBS containing the MitoSOX reagent (2.5 μM) and incubated for 10 min at 37 °C with light protection. The fluorescence was measured at 510/580 nm using an M200 Infinite Pro plate reader (Tecan, Männedorf, Switzerland). The results were normalized to the protein content using the Pierce BCA Protein Assay kit (Thermo Fisher Scientific, Basel, Switzerland).

### 4.9. Annexin V/Propidium Iodide Staining

Apoptosis (early and late apoptosis/necrosis) was determined using the Alexa Fluor^®^ 488 annexin V/propidium iodide (PI) staining kit, according to the manufacturer’s protocol (Vybrant TM Apoptosis Assay Kit #2) (Gibco Life Technologies, Paisley, UK) [63] and was followed by flow cytometric acquisition using a Cytoflex cytometer (Beckman Coulter, Indianapolis, IN, USA). Briefly, undifferentiated and differentiated SH-SY5Y cells were seeded into a 24-well plates and exposed to amphetamine, 4-FA, methcathinone, 4-FMC, or 4-CMC (1000 μM and 2000 μM), or PCA (100 μM, 200 μM, and 500 μM) for 6 h at 37 °C. This shorter incubation time (6 h versus 24 h) was selected to focus on the early apoptosis phase rather than on necrosis. Following incubation, the cells were detached and transferred into a V-well plate. The cells were pelleted and washed twice with PBS, then 100 μL of 1X annexin-binding buffer containing 5 μL of Alexa Fluor^®^ 488 annexin V, 1 μL of PI (100 μg/mL), and 0.5 μL of anti-CD29-APC (clone TS2/16) (BioLegend, San Diego, CA, USA) was added to each well. CD29 is a member of the integrin family which is expressed on the plasma membrane of undifferentiated and differentiated SH-SY5Y cells. Thereafter, the plate was incubated at 4 °C for 15 min with light protection. For the flow cytometry gating strategy, singlets were first identified by a forward scatter area (FSC-A) and forward scatter height (FSC-H) gate, and then by an FSC-A and a side scatter area (SSC-A) gate. Intact SH-SY5Y cells were distinguished from cell debris through staining with anti-CD29-APC in a FL-5 and an SCC-A gate. Samples were analyzed using FlowJo software (Tree Star, Ashland, OR, USA).

### 4.10. Western Blotting

Undifferentiated and differentiated SH-SY5Y cells were grown into a 6-well plate and treated with 4-CMC (500 μM, 1000 μM, and 2000 μM) for 6 h. Amiodarone 50 μM and MDMA 1000 μM were used as negative controls [62,64]. After treatment, cells were washed twice with cold PBS buffer (pH 7.4) and incubated with 70 μL of cell lysis buffer (radioimmunoprecipitation assay (RIPA) buffer supplemented with complete protease inhibitor (Roche Diagnostics, Mannheim, Germany)) for 15 min on ice. Then, the cell lysates were collected into Eppendorf tubes and centrifuged at 1600 *g* for 20 min. The supernatants were collected and the protein concentration was quantified using the Pierce BCA Pierce Protein Assay kit (Thermo Fisher Scientific, Basel, Switzerland). The protein extracts (18 μg per lane) were loaded, separated by SDS/PAGE using 4–12% NuPAGE Bis-Tris gels (Invitrogen, Basel, Switzerland), and then transferred to nitrocellulose membranes by the Trans-Blot Turbo Blotting System (Bio-Rad, Cressier, Switzerland). After protein transfer, membranes were first incubated with blocking buffer (5% non-fat dry milk in PBS containing 0.1% Tween-20 (Sigma-Aldrich, Buchs, Switzerland)) for 1 h at RT, and then incubated overnight at 4 °C with the following primary antibodies diluted in blocking solution: anti-cleaved caspase 3 (diluted 1:500, ab32042, RRID:AB_725947, Abcam, Cambridge, UK), anti-caspase 3 (diluted 1:500, 8G10, RRID:AB_2069872, Cell Signaling Technology, Danvers, USA), anti-caspase 9 (diluted 1:2000, ab52298, RRID:AB_868689, Abcam, Cambridge, UK) for full and cleaved forms, and anti-glyceraldehyde 3-phosphate dehydrogenase (GAPDH) (sc-365062, RRID:AB_10847862, Santa Cruz Biotechnology, Dallas, TX, USA). The amount of GAPDH was used to correct for different loading. Thereafter, the nitrocellulose membranes were washed three times with PBS and incubated with secondary antibodies (Santa Cruz Biotechnology, Dallas, TX, USA) diluted at 1:2000 in blocking solution for 1 h at RT. Membranes were then washed, and protein bands were developed using the ClarityTM Western ECL Substrate (Bio-Rad Laboratories, San Diego, CA, USA). Protein expression was quantified using the Fusion Pulse TS device (Vilber Lourmat, Oberschwaben, Germany).

### 4.11. Statistics

GraphPad Prism 8.3.0 (GraphPad Software, La Jolla, CA, USA) was used for all statistical analyses. The data are presented as the mean ± standard error mean (SEM) of at least three independent experiments. Statistical differences between the DMSO 0.1% control group and test drugs were calculated with one-way ANOVA followed by the Dunnett’s test. A *p*-value < 0.05 was considered to indicate a statistically significant difference.

## 5. Conclusions

In conclusion, *para*-halogenation of amphetamine and methcathinone increases mitochondrial toxicity associated with these drugs. The *para*-chlorinated and fluorinated forms were toxic on undifferentiated and differentiated SH-SY5Y cells in a concentration-dependent fashion, whereby the amphetamine derivatives showed higher toxicity than the methcathinone counterparts. Moreover, differentiated SH-SY5Y cells were less susceptible to the toxic effects of the compounds investigated, possibly due to stronger antioxidative capacity. Although the cytotoxic concentrations were higher than those needed for pharmacological activity, mitochondrial dysfunction may represent a major mechanism for neural toxicity associated with these compounds.

## Figures and Tables

**Figure 1 ijms-21-02841-f001:**
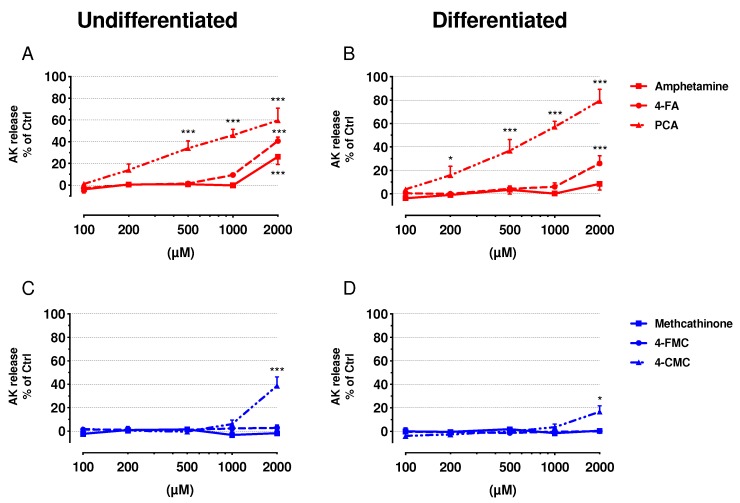
Plasma membrane integrity in undifferentiated and differentiated SH-SY5Y cells. (**A**) Undifferentiated and (**B**) differentiated SH-SY5Y cells were exposed to amphetamine, 4-FA and PCA for 24 h. (**C**) Undifferentiated and (**D**) differentiated SH-SY5Y cells were exposed to methcathinone, 4-FMC and 4-CMC for 24 h. Data are expressed as percentage of release of adenylate kinase (AK) compared to control (controls: DMSO 0.1% set as 0%, Triton X set as 100%). Data are expressed as mean ± SEM of eight independent experiments. Statistical differences were calculated with one-way ANOVA followed by the Dunnett’s test, (*) *p* ≤ 0.05 and (***) *p* ≤ 0.001 versus 0.1% DMSO control.

**Figure 2 ijms-21-02841-f002:**
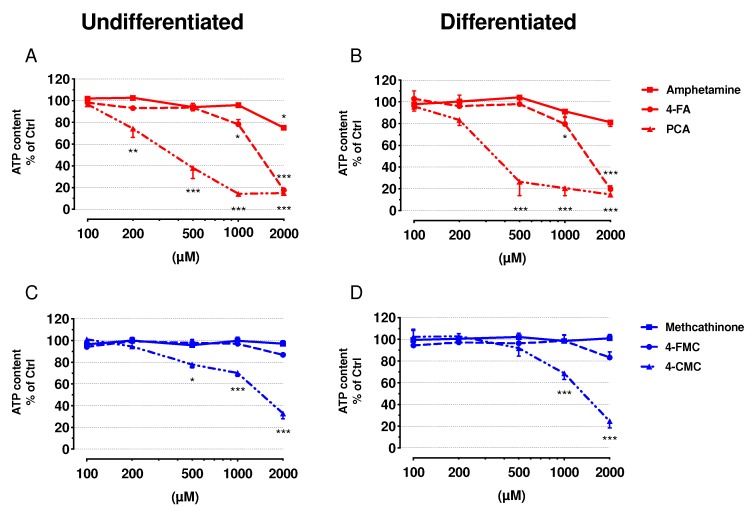
Intracellular ATP content in undifferentiated and differentiated SH-SY5Y cells. (**A**) Undifferentiated and (**B**) differentiated SH-SY5Y cells were exposed to amphetamine, 4-FA and PCA for 24 h. (**C**) Undifferentiated and (**D**) differentiated SH-SY5Y cells were exposed to methcathinone, 4-FMC and 4-CMC for 24 h. Data are expressed as percentage of ATP content compared to control (controls: DMSO 0.1% set as 100%, Triton X set as 0%. Data are expressed as mean ± SEM of eight independent experiments. Statistical differences were calculated with one-way ANOVA followed by the Dunnett’s test, (*) *p* ≤ 0.05, (**) *p* ≤ 0.01 and (***) *p* ≤ 0.001 versus 0.1% DMSO control.

**Figure 3 ijms-21-02841-f003:**
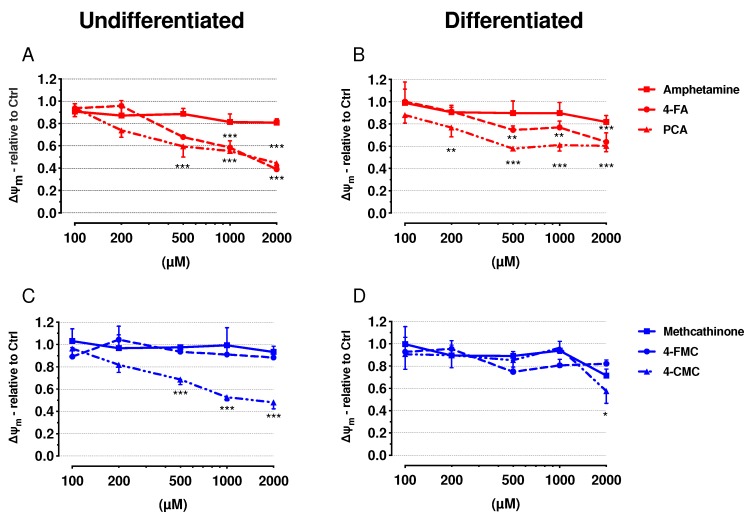
Mitochondrial membrane potential in undifferentiated and differentiated SH-SY5Y cells. Mitochondrial membrane potential was measured by JC-1 staining after exposure to drugs in undifferentiated and differentiated SH-SY5Y cells for 24 h. (**A**) Undifferentiated and (**B**) differentiated SH-SY5Y cells were exposed to amphetamine, 4-FA and PCA for 24 h. (**C**) Undifferentiated and (**D**) differentiated SH-SY5Y cells were exposed to methcathinone, 4-FMC and 4-CMC for 24 h. Data are expressed as mean ± SEM of five independent experiments. Statistical differences were calculated with one-way ANOVA followed by the Dunnett’s test, (*) *p* ≤ 0.05, (**) *p* ≤ 0.01 and (***) *p* ≤ 0.001 versus 0.1% DMSO control.

**Figure 4 ijms-21-02841-f004:**
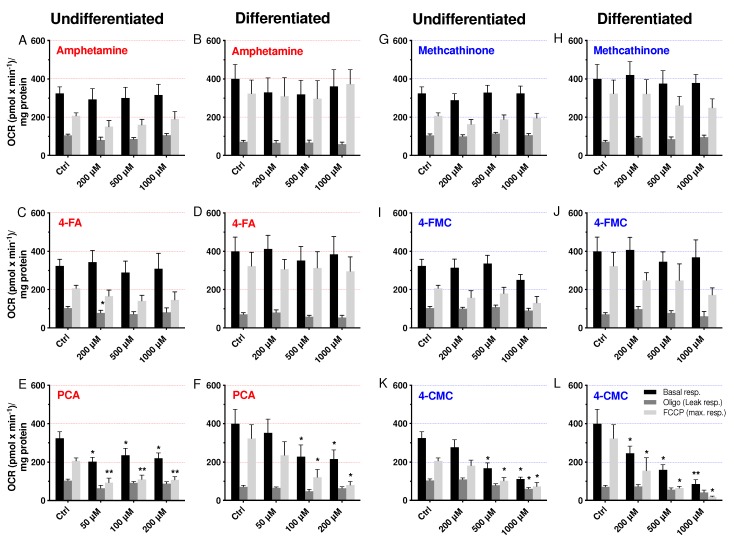
Oxygen consumption by undifferentiated and differentiated SH-SY5Y cells expressed as basal, leak, and maximal respiration. Undifferentiated and differentiated SH-SY5Y cells were exposed to (**A**,**B**) amphetamine, (**C**,**D**) 4-FA, (**E**,**F**) PCA, (**G**,**H**) methcathinone, (**I**,**J**) 4-FMC, and (**K**,**L**) 4-CMC for 24 h. Data are expressed as mean ± SEM of seven independent experiments. Statistical differences were calculated with one-way ANOVA followed by the Dunnett’s test, (*) *p* ≤ 0.05, (**) *p* ≤ 0.01 versus 0.1% DMSO control.

**Figure 5 ijms-21-02841-f005:**
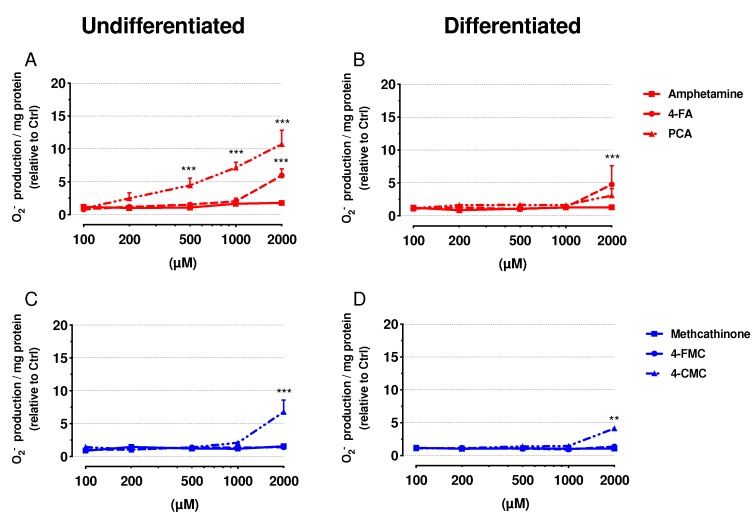
Mitochondrial superoxide accumulation was measured in undifferentiated and differentiated SH-SY5Y cells after exposure to drugs for 24 h. (**A**) Undifferentiated and (**B**) differentiated SH-SY5Y cells were exposed to amphetamine, 4-FA and PCA for 24 h. (**C**) Undifferentiated and (**D**) differentiated SH-SY5Y cells were exposed to methcathinone, 4-FMC and 4-CMC for 24 h. Data are expressed as mean ± SEM of five independent experiments. Statistical differences were calculated with one-way ANOVA followed by the Dunnett’s test, (**) *p* ≤ 0.01 and (***) *p* ≤ 0.001 versus 0.1% DMSO control.

**Figure 6 ijms-21-02841-f006:**
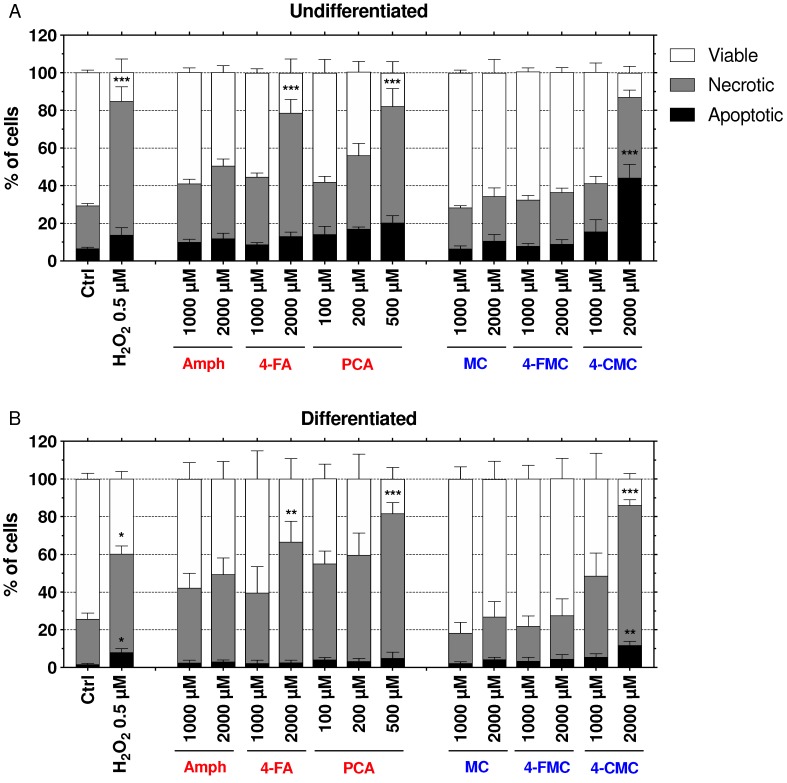
Mechanisms of cell death of undifferentiated and differentiated SH-SY5Y cells exposed to toxicants for 6 h. (**A**) Undifferentiated and (**B**) differentiated SH-SY5Y cells were exposed to amphetamine (Amph), 4-FA, PCA, methcathinone (MC), 4-FMC, and 4-CMC for 6 h. Data is expressed as mean ± SEM of six independent experiments. Statistical differences were calculated with one-way ANOVA followed by the Dunnett’s test, (*) *p* ≤ 0.05, (**) *p* ≤ 0.01 and (***) *p* ≤ 0.001 versus 0.1% DMSO control.

**Figure 7 ijms-21-02841-f007:**
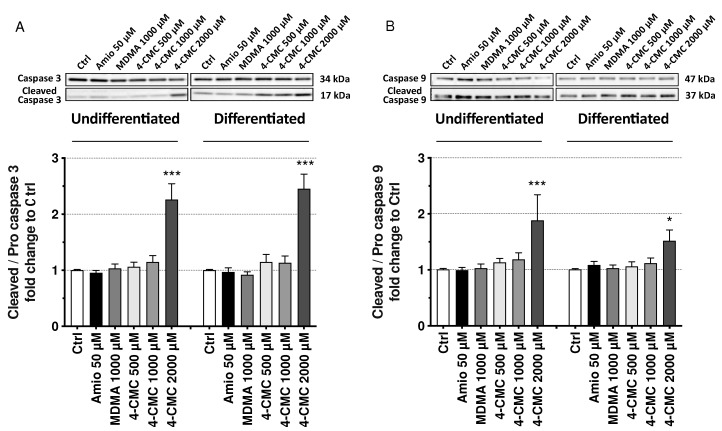
Effects on markers of apoptosis in undifferentiated and differentiated SH-SY5Y cells exposed to drugs for 6 h. (**A**) Activation of caspase 3 and (**B**) activation of caspase 9. Data are expressed as mean ± SEM of eight independent experiments for each drug concentration. Statistical differences were calculated with one-way ANOVA followed by the Dunnett’s test, (*) *p* ≤ 0.05, and (***) *p* ≤ 0.001 versus 0.1% DMSO control.

**Table 1 ijms-21-02841-t001:** Quantification (IC_50_) of membrane toxicity (MT) and ATP depletion (ATP) by *para*-halogenated amphetamine (Amph) and methcathinone (MC) derivatives in undifferentiated (und), and differentiated (diff) SH-SY5Y cells.

Drug	Amph		4-FA		PCA		MC		4-FMC		4-CMC	
**Cell**	und	diff	und	diff	und	diff	und	diff	und	diff	und	diff
**MT (IC_50_)** **[mM]**	>2	>2	>2	>2	1.17	0.77	>2	>2	>2	>2	>2	>2
**ATP (IC_50_)** **[mM]**	>2	>2	1.44	1.41	0.42	0.39	>2	>2	>2	>2	1.43	1.33

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
