# Peer review of "Para-Halogenation of Amphetamine and Methcathinone Increases the Mitochondrial Toxicity in Undifferentiated and Differentiated SH-SY5Y Cells"

_ijms, 2020, doi:10.3390/ijms21082841_

Round 1

Reviewer 1 Report

The investigators showed previously that para-halogenated derivatives of amphetamine and methcathinone disrupts the mitochondrial respiratory chain, impairs the mitochondrial function and contributes to hepatotoxicity. In the present study, using a differentiated and undifferentiated SH-SY5Y cell model, the authors investigated the effect of amphetamine, methcathinone and their para-halogenated derivatives on neurotoxicity and mitochondrial function. They found that para-halogenation of amphetamine and methcathinone impairs mitochondrial function in both differentiated and undifferentiated cells by ATP depletion, increased ROS production, and results in neurotoxicity.

The manuscript is well written with good description of all the methods, results and statistics. Overall, the paper provides a clear and convincing view of the neurotoxic effect of amphetamine, methcathinone and their derivatives due to impairment of the mitochondrial function.

I have only a few comments:

  1. Why only 4CMC was used to study the mechanisms of cell death. I don’t clearly understand the reason for the selection of only this derivative for the study, why Caspase 3 and 9 expressions were not checked in PCA treated cells. Can you please clarify this better for the reader? I’m referring to section 2.6: mechanisms of cell death.
  2. Did authors checked the effect of amphetamine (Amph) and methcathinone (MC) derivatives on the electron transport chain complexes activity. This could further provide the mechanistic insight for enhanced ROS production and ATP depletion and would considerably strengthen the manuscript.

Author Response

  1. Why only 4CMC was used to study the mechanisms of cell death. I don’t clearly understand the reason for the selection of only this derivative for the study, why Caspase 3 and 9 expressions were not checked in PCA treated cells. Can you please clarify this better for the reader? I’m referring to section 2.6: mechanisms of cell death.

Answer: As shown in Figure 6 and described on page 7 of the manuscript, only 4-CMC induced apoptosis, whereas PCA and 4-FA induced necrosis at the time point investigated (incubation for 6 h). Therefore, we chose 4-CMC to investigate the effect on caspases. In retrospect, it would have been good to include also PCA as suggested by the Reviewer, in order to see the difference to 4-CMC.

We have modified the text on page 7 to make clear, why we included 4-CMC for studying the effects on caspases and have improved the description of the results shown Figure 6 in the text.

Furthermore, we have observed that we have included a Figure 6, which had not the correct results for 4-CMC-associated cell damage in differentiated cells. We have therefore replaced Figure 6.

  1. Did authors checked the effect of amphetamine (Amph) and methcathinone (MC) derivatives on the electron transport chain complexes activity. This could further provide the mechanistic insight for enhanced ROS production and ATP depletion and would considerably strengthen the manuscript.

Answer: In the current study, we have shown that PCA and 4-CMC impair oxygen uptake by SH-SY5Y cells under basal conditions and in the presence of the uncoupler FCCP. Since >90% of the cellular oxygen is used by mitochondria and since the oxygen within mitochondria is used for the function of the electron transport chain, these results suggest impaired function of the mitochondrial electron transport chain in SH-SY5Y cells. The results do however not allow to conclude which complexes(s) of the electron transport chain is or are affected. For that measurements with different substrates or direct determination of the activity of individual complexes would have been necessary. We have done that in a previous study using similar compounds in C2C12 cells (a murine skeletal muscle cell line) (Int J Mol Sci. 2019;20:E1561. doi: 10.3390/ijms20071561). In this previous study, the amphetamine-like compound MDMA inhibited complex I and III, and the cathinones α-PVP complex I and naphyrone complex II. Complex I and III are targets of these drugs; their inhibition would explain the findings in the current study.

We have made an addition in the discussion section (page 10) stating that inhibition of complex I and/or III, that was found in a previous investigation in another cell-line could explain the findings in the current study.

Reviewer 2 Report

Dear Editor,

The article “Para-halogenation of amphetamine and 2 methcathinone increases the mitochondrial toxicity in undifferentiated and differentiated SH-SY5Y cells” experimentally proved that amphetamine and 2 methcathinone, which is generally used as a psychoactive substance, have toxic effects on mitochondria which is the major driving source for cellular energy production. Article has been scientifically well written and would have a good implication in the respect field. However, I have few significant points (written in the authors section) in the mechanistic approach which authors presented in this article. Providing further information would significantly improve the quality of this manuscript.

Thanks

Author Response

The article “Para-halogenation of amphetamine and 2 methcathinone increases the mitochondrial toxicity in undifferentiated and differentiated SH-SY5Y cells” experimentally proved that amphetamine and 2 methcathinone, which is generally used as a psychoactive substance, have toxic effects on mitochondria which is the major driving source for cellular energy production. Article has been scientifically well written and would have a good implication in the respect field.

However, I have few significant points (written in the authors section) in the mechanistic approach which authors presented in this article.

Providing further information would significantly improve the quality of this manuscript.

Answer: Unfortunately, we did not find the remarks of the Reviewer. In the questions asked to the Reviewer, the Reviewer indicated that the English language should be checked. We went over the entire manuscript and corrected errors as well as we could.

Round 2

Reviewer 2 Report

Dear Authors,

I am not sure, why you have not received my comments previously. These are again included below;

The article has beautifully provided experiments, and logically concluded that mitochondrial dysfunction caused by amphetamine and 2 methcathinone, and its halogenated derivatives is by the induction of apoptosis pathways. Mechanistically, western blot data of cleaved caspase 3 and total caspase 3 has been provided. I still think that other mechanisms may also be activated by amphetamine and 2 methcathinone, which has not been discussed in this manuscript. My major points are listed below-

  1. Authors have shown that ROS has been activated by these psychoactive drugs. Did authors check the level of antioxidant genes or proteins in these experiments?

  1. Authors have shown that in his previous findings they checked these psychoactive drugs on the muscle C2C12 cells. Authors did not mention in the discussion, how both studies mimic the findings. Please include this part in discussion.

  1. Using Seahorse, authors have shown that these compounds decrease the mitochondrial respiration. Did authors check the ECAR data in Seahorse? That may be important because authors have shown that ATP production have also been decreased after drugs treatment, and glycolysis may also have some contribution in ATP generation.

  1. These drugs impair the mitochondrial function, and it may possible that bulk autophagy or Mitophagy may have activated by these drugs in these cells to remove dysfunctional mitochondria. Did authors check the autophagy markers at mRNA or protein level?

  1. Did author check any mitophagy pathways (PINK-PARKIN, or BNIP3 dependent) in these cells after drug treatment.

Author Response

  1. Authors have shown that ROS has been activated by these psychoactive drugs. Did authors check the level of antioxidant genes or proteins in these experiments?

Answer: We didn’t do that in this study. Taking into account that ROS accumulation and impairment of the cellular oxygen uptake (indicating damage to the mitochondrial electron transport chain) appeared at similar concentrations, mitochondrial dysfunction (mainly of complex I and III; please, read also our answer to the second question of Reviewer 1) is most probably the reason for increased ROS production. Within the mitochondria, mainly superoxide is produced and degraded by SOD2 to H2O2, which can leave the mitochondria and oxidize KEAP1 with subsequent stabilization and activation of Nrf2. Nrf2 is a nuclear transcription factor that increases the expression of many proteins involved in antioxidative defense, for instance also for SOD2.

We have shown this sequence in previous studies in other cell types, e.g. in HepG2 cells (Free Radic Biol Med 2020;152:216-226). It is a well-conserved and important defense mechanism of most cells. We therefore assume that this is also true for SH-SY5Y cells.

Unfortunately, we cannot perform new experiments at this moment due to restricted access to the lab because of the COVID-19 situation. We, however, describe this mechanism as a possible consequence of mitochondrial superoxide accumulation on page 10 of the manuscript.

  1. Authors have shown that in his previous findings they checked these psychoactive drugs on the muscle C2C12 cells. Authors did not mention in the discussion, how both studies mimic the findings. Please include this part in discussion.

Answer: Thank you for this notion, we fully agree. We have included these experiments into discussion (page 10) of the manuscript. Please, read also our answer to question 2 of Reviewer 1.

  1. Using Seahorse, authors have shown that these compounds decrease the mitochondrial respiration. Did authors check the ECAR data in Seahorse? That may be important because authors have shown that ATP production have also been decreased after drugs treatment, and glycolysis may also have some contribution in ATP generation.

Answer: This is correct; since we used glucose in the buffer of the Seahorse experiments, the cells were able to generate ATP by glycolysis (we state that on page 9 of the manuscript). Unfortunately, we did not record the ECAR data, which we now regret. On the other hand, a mitochondrial mechanism is present without any doubt: the cellular oxygen consumption was impaired and the mitochondrial membrane potential dropped at similar concentrations as we observed the drop in the cellular ATP pool. Furthermore, the drop in the cellular ATP pool appeared earlier (at lower concentrations) than membrane damage, also favoring a mitochondrial mechanism.

  1. These drugs impair the mitochondrial function, and it may possible that bulk autophagy or Mitophagy may have activated by these drugs in these cells to remove dysfunctional mitochondria. Did authors check the autophagy markers at mRNA or protein level?

Answer: Autophagy can be considered as a cellular defense mechanism to avoid apoptosis. Since we observed apoptosis, it can be assumed that there was also mitophagy to remove damaged proteins and organelles such as mitochondria in order to prevent apoptosis. We performed an additional study that has been accepted by Cells (Zhou, X., et al., Hyperthermia increases neurotoxicity associated with novel methcathinones. Cells, 2020. 9:in press. Please, see attachment for this article). In this study we confirmed the appearance of autophagy in SH-SY5Y cells treated with 4-CMC. We describe this result on page 10 of the manuscript and cite the article.

  1. Did author check any mitophagy pathways (PINK-PARKIN, or BNIP3 dependent) in these cells after drug treatment.

Answer: In order to detect autophagy, we determined the LC3 II/LC3 I ratio by western blotting (Zhou, X., et al., Hyperthermia increases neurotoxicity associated with novel methcathinones. Cells, 2020. 9: p. in press). In addition, we visualized and quantified acidic vesicular organelles by staining with acridine orange using fluorescence microscopy. Using these techniques, we could demonstrate that 4-CMC stimulates autophagy in SH-SY5Y cells. We mention the results on page 10 of the manuscript and cite the paper.

Round 3

Reviewer 2 Report

My concerns has been resolved. 

Thanks!